# PHASE-GUIDED PERCEPTUAL ALIGNMENT FOR MULTI-SOURCE MULTI-MODAL DOMAIN ADAPTATION

## ABSTRACT

Multi-Source Multi-Modal Domain Adaptation ($MSM^2DA$) is a method that leverages data from multiple sources and modalities to train machine learning models capable of generalizing well across various domains. Existing $MSM^2DA$ methods mostly use structural semantic alignment by visual data to enhance the correlation between different modality data, while neglecting the low-frequency perceptual shifts in visual data that hinder cross-modal fusion. However, visual data are particularly sensitive to domain shifts including low-level semantics such as style and illumination variations. To handle this problem, we propose Phase-guided Perceptual Alignment (PGPA) to align the visual styles by transferring low-frequency spectral components from target to source images while preserving high-frequency semantic structures. Specifically, PGPA decomposes images into amplitude and phase spectra in the Fourier domain, where the amplitude captures style-related low-level statistics and the phase retains high-level structural semantics. By selectively blending the amplitude of the target image with the phase of the source image, our method improve diversity and ensures domain-invariant style adaptation without distorting critical semantic details. Furthermore, we provide a bound proof that formalizes the effectiveness of our approach, demonstrating that PGPA guarantees improved cross-domain generalization within a specified bound and ensuring theoretical validity. Extensive experiments demonstrate that our approach significantly improves cross-domain generalization tasks.

## 1 INTRODUCTION

Unsupervised domain adaptation (UDA) Wilson & Cook (2020) aims to transfer knowledge from a labeled source domain to an unlabeled target domain, assuming task consistency but distributional differences. However, conventional UDA methods often consider only a single source and a single modality, which does not reflect the complexity of real-world data.

To address this, multi-source domain adaptation (MSDA) Sun et al. (2015) extends UDA by utilizing multiple labeled source domains. It improves generalization by mitigating inter-source distribution gaps through techniques such as domain-specific encoders, source weighting, and shared latent space learning. However, most existing MSDA approaches are restricted to unimodal settings and fail to capture multi-modal interactions. Multi-modal domain adaptation (MMDA) Hu et al. (2023) focuses



(a) **Source Domain Image**     (b) **Target Domain Image**     (c) **Source Image in Target style**

Figure 1: Effectiveness of proposed Phase-guided Perceptual Alignment. The source image (a) is transformed by our proposed PGPA from the target image (b), resulting in an aligned image (c) that reduced perceptual domain gap and preserved semantics.

on aligning different modalities such as image and text under domain shifts. Traditional methods often rely on early or late fusion and use adversarial learning or distribution matching. However, these methods usually assume a single source domain and overlook challenges from source heterogeneity. As a result, they experience performance degradation when extended to multi-source settings.

To address this, multi-source multi-modal domain adaptation (MSM$^2$DA) has recently gained attention Zhao et al. (2025). MSM$^2$DA aims to train a generalized model using multiple labeled source domains with diverse modalities to achieve strong performance in an unlabeled target domain. This approach is more aligned with practical cross-modal applications where both domain generalization and modality fusion are crucial. In the MSM$^2$DA setting, the visual modality plays a critical role in model performance. Compared to other modalities, visual data typically contain more structural information and act as an anchor for multi-modal semantic alignment. However, the visual modality is highly sensitive to domain differences. For example, images from different sources often differ significantly in visual style, even in similar tasks. These perceptual differences, although they do not affect semantic content, introduce large distribution shifts that hinder feature learning. This results in inconsistencies during the fusion stage, creating a bottleneck for multi-modal alignment. Therefore, reducing domain shift in the visual modality is essential for stable fusion and improved transfer performance.

To tackle this challenge, we introduce Phase-guided Perceptual Alignment (PGPA), a method that leverages Fourier transform for frequency-based alignment in the visual modality. The key idea behind PGPA is to enhance diversity by shifting alignment to the frequency domain, specifically targeting the low-frequency components. By doing so, PGPA reduces perceptual conflicts between modalities and mitigates the impact of domain shifts. PGPA works by injecting low-frequency style information from the target domain into the source domain images before they enter the multi-modal model. This process narrows the perceptual gap in visual style while maintaining the integrity of semantic structures. As illustrated in Figure 1, PGPA effectively transfers the target domain's visual style to the source image through low-frequency alignment, producing an image with reduced domain discrepancy and preserved semantic content. Additionally, we provide a theoretical proof that demonstrates PGPA's ability to reduce error. Our approach enhances the diversity of multi-source multi-modal data through Fourier transform-based alignment, bridging both modality and domain gaps. PGPA is training-free, architecture-independent, and can be seamlessly integrated into any MSM$^2$DA framework. By performing pixel-level alignment in the visual stream, PGPA offers a stable foundation for multi-modal fusion and domain adaptation. Overall, the main contribution of this paper can be summarized as follows:

- We propose a novel method called PGPA which reduces perceptual conflicts between modalities and mitigates the impact of domain shifts by aligning the low-frequency components, which improve stability in multi-modal fusion and domain adaptation.

- We provide a theoretical proof that demonstrates PGPA's ability to reduce error and improve alignment accuracy. This formal validation strengthens the theoretical foundation of PGPA, showing its effectiveness in addressing domain shift and enhancing the transfer performance in MSM$^2$DA.

- Our method achieves state-of-the-art performance on MSM$^2$DA benchmarks such as aesthetics assessment and sentiment analysis, effectively extending previously successful MSDA approaches to the MSM$^2$DA setting.

## 2 RELATED WORK

Unsupervised domain adaptation bridges the gap between a labeled source domain and an unlabeled target domain by reducing distribution shifts Shrivastava et al. (2017); Mekhazni et al. (2020); Huang & Liu (2021); Li et al. (2024). Although extensive research has been devoted to single-source, single-modal settings, real-world applications often involve multiple heterogeneous sources and diverse modalities. This complexity gives rise to more challenging scenarios, including multi-source domain adaptation, multi-modal domain adaptation, and multi-source multi-modal domain adaptation. Multi-source domain adaptation seeks to leverage labeled data from multiple source domains to enhance generalization on an unlabeled target domain. Compared to single-source settings, MSDA address not only source-target discrepancies but also source-source inconsistencies. Existing methods are mainly classified into three categories: aligning domain distributions via adversarial

learning or moment-based metrics Zhao et al. (2020); Gao et al. (2024), generating domain-invariant representations through intermediate feature space Zhao et al. (2019); Lin et al. (2020), or refining classifiers to reduce inter-domain variance and inter-class ambiguity Zhu et al. (2019); Karisani (2022). Although these approaches have shown effectiveness, they often assume a single modality and struggle when modality gaps are coupled with domain shifts. Multi-modal domain adaptation addresses the challenge of transferring knowledge across different modalities within a single domain. The existing work mainly focuses on exploring early and late fusion strategies, with alignment occurring at the modality-specific feature level or after fusion Munro & Damen (2020); Jaritz et al. (2020); Peng et al. (2021). However, most existing MMDA methods focus on single-source scenarios and lack the capacity to unravel complex interactions between source diversity and modality heterogeneity. To bridge domain and modality discrepancies, M2CAN Zhao et al. (2025) introduces a unified framework that combines contrastive and adversarial learning for joint alignment. It performs multilevel alignment across feature and prediction spaces through cross-modal contrastive learning, cross-domain contrastive alignment, and adversarial objectives. However, M2CAN mainly targets high-level semantic features and overlooks low-frequency perceptual style shifts in the visual modality. These seemingly irrelevant variations can disrupt multimodal fusion and hinder cross-modal interaction. To address this, we propose a model-agnostic, plug-and-play alignment strategy that aligns visual styles at the input level. By shifting the alignment perspective to the perceptual layer, our method facilitates more stable multimodal fusion and cross-domain generalization.

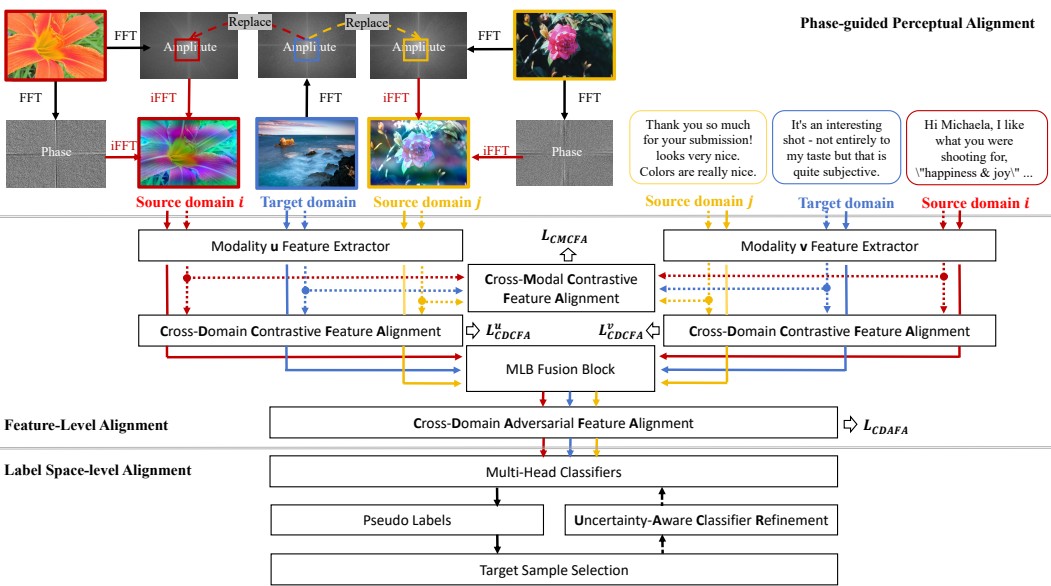

Figure 2: Overview of proposed PGPA in the MSM$^2$DA framework. PGPA selectively replaces the low-frequency amplitude of source images with that from the target domain in the Fourier space, preserving semantic structures while reducing style discrepancy. The aligned images and other modalities are then fed into a multi-source multi-modal adaptation framework consisting of four modules: CMCFA, CDCFA and CDAFA for feature-level alignment, and UACR for label space-level alignment.

## 3 METHOD

### 3.1 PROBLEM STATEMENT

Multi-source Multi-modal Domain Adaptation is considered under the covariate shift assumption. Let $\mathcal{S} = \{\mathcal{S}_i\}_{i=1}^N$ denote $N$ labeled source domains, and let $\mathcal{T}$ denote the target domain, which contains only unlabeled samples. Each source domain $\mathcal{S}_i$ consists of examples drawn from a joint distribution $p^{(\mathcal{S}_i)}(x_1, x_2, \ldots, x_M, y)$ over $M$ modalities and the label space $\mathcal{Y}$, with input space $\mathcal{X}_1 \times \cdots \times \mathcal{X}_M$. Although all domains share the same input and output spaces, their joint distributions

differ, and there may also be a distributional gap among different source domains. Our goal is to learn a multi-modal classifier $f : \mathcal{X}_1 \times \cdots \times \mathcal{X}_M \to \mathcal{Y}$ that can generalize to the target domain using only labeled data from the source domains.

## 3.2 PHASE-GUIDED PERCEPTUAL ALIGNMENT

In multi-source multi-modal domain adaptation tasks, each source domain may involve multiple modalities. For a given modality, its perceptual characteristics and representational forms often vary across different domains, leading to inconsistent behaviors during domain transfer. This issue is especially pronounced in the visual modality, as cross-domain discrepancies in image style, illumination, and background textures often emerge as key bottlenecks that hinder transfer performance. While these low-level perceptual variations do not alter the semantic content of an image, they can still disrupt the early stages of representation learning. To better understand and mitigate their impact, we consider the frequency-domain perspective. From this point of view, these domain-induced differences are primarily encoded in the low-frequency components of the image spectrum, corresponding to global attributes such as tone, brightness, and background layout. In contrast, the high-frequency components tend to preserve fine-grained structures like edges and textures, which are more semantically informative.

Motivated by the above observation, we target the low-frequency statistical discrepancies in the visual modality by proposing a Phase-guided Perceptual Alignment (PGPA) method. PGPA modifies the low-frequency amplitude spectrum of source images to match that of target images, while retaining the high-frequency structure that carries semantic information. This operation reduces perceptual mismatches across domains without altering semantic content, thereby enhancing visual consistency during domain adaptation. Given that such low-level perceptual variations are unique to the visual modality, we apply frequency-domain alignment exclusively to the image modality, while leaving the remaining modalities unchanged. Formally, given source and target image modalities $z_s^{(i)}, z_t \in \mathbb{R}^{H \times W \times C}$, we compute Fourier transforms $\mathcal{F}$ for a single-channel image $z$ as:

$$\mathcal{F}(z)(h', w') = \sum_{h=0}^{H-1} \sum_{w=0}^{W-1} z(h, w) \cdot e^{-j2\pi \left( \frac{hh'}{H} + \frac{ww'}{W} \right)} \tag{1}$$

The Fourier transform $\mathcal{F}$ consists of an amplitude $\mathcal{F}_A(z)$ and a phase $\mathcal{F}_P(z)$, which capture frequency magnitude and structural information, respectively.

To selectively replace domain-specific style statistics while preserving semantic structure, we introduce a low-frequency mask $M_\beta(h, w) \in \{0, 1\}$, which is applied to the amplitude spectrum. Specifically, the mask takes the value 1 within a centered rectangular region of size $(2\beta H) \times (2\beta W)$, and 0 elsewhere. Formally, the low-frequency mask is defined as follow:

$$M_\beta(h, w) = \mathbf{1}_{(h,w) \in [-\beta H : \beta H, \; -\beta W : \beta W]}, \tag{2}$$

where $\beta \in (0, 1)$ controls the proportion of low-frequency components to be transferred. The transformed source image $\tilde{z}_s^{(i)}$ is then constructed by blending the low-frequency amplitude from a randomly sampled target image $z_t$ with the original source amplitude outside the masked region, while retaining the source phase:

$$\tilde{z}_s^{(i)} = \mathcal{F}^{-1} \left( M_\beta \cdot \mathcal{F}_A(z_t) + (1 - M_\beta) \cdot \mathcal{F}_A(z_s^{(i)}), \; \mathcal{F}_P(z_s^{(i)}) \right) \tag{3}$$

$z_s^{(i)}$ visually resembles the target domain in style while preserving its semantic content. We independently process each source domain to obtain an aligned set as follow:

$$\widetilde{\mathcal{S}}_i = \left\{ \left( \tilde{z}_s^{(i)}, y_s^{(i)} \right) \mid z_s^{(i)} \sim \mathcal{S}_i, \; z_t \sim \mathcal{T} \right\} \tag{4}$$

The final training set is obtained by merging all adapted source domains $\widetilde{\mathcal{S}} = \bigcup_{i=1}^{N} \widetilde{\mathcal{S}}_i$. Then $\widetilde{\mathcal{S}}$ and $\mathcal{T}$ are fed into a multi-modal domain adaptation framework. It is important to note that PGPA is independent of downstream network architectures, and can be used as a standalone input-level perceptual alignment strategy for the image modality. Its primary objective is to mitigate low-frequency perceptual discrepancies across domains. By preserving semantic content while harmonizing perceptual styles, PGPA improves cross-domain visual consistency and facilitates more effective feature extraction and multi-modal fusion.

### 3.3 Multi-source Multi-modal Contrastive Adversarial Network

**Cross-modal Contrastive Feature Alignment (CMCFA).** CMCFA reduces representational gaps between different modalities within a single source domain. For modalities $u$ and $v$, let $X_u$ and $X_v$ denote batches of original features, and $X'_u$ and $X'_v$ denote their augmented versions. The base contrastive loss between modalities $u$ and $v$ is defined as:

$$\mathcal{L}_{\text{CMCFA}}^{uv} = -\frac{1}{n} \cdot \mathbf{1}^\top \cdot \log \left[ \frac{e^{\mathbb{I} \circ T} + e^{\mathbb{I} \circ T'} + e^{\mathbb{I}' \circ T} + e^{\mathbb{I}' \circ T'}}{\mathbf{1}^\top \cdot \left( e^{\mathbb{I} \cdot T^\top} + e^{\mathbb{I} \cdot T'^\top} + e^{\mathbb{I}' \cdot T^\top} + e^{\mathbb{I}' \cdot T'^\top} \right) \cdot \mathbf{1}} \right] \tag{5}$$

where $\mathbb{I} = X_u$, $\mathbb{I}' = X'_u$, $T = X_v$, $T' = X'_v$; $\circ$ denotes the Hadamard product. To avoid forced alignment of mismatched modalities, CMCFA estimate modality matching using KL-divergence between predictions of modality-specific classifiers. For classifiers $F_u$ and $F_v$ for modalities $u$ and $v$, the mismatch variance is:

$$\text{Var}^{uv} = \text{KL}\left( F_u(X_u|\theta_u), F_v(X_v|\theta_v) \right) \tag{6}$$

where a larger $\text{Var}^{uv}$ indicates lower matching. The final CMCFA loss aggregates all modality pairs as follow:

$$\mathcal{L}_{\text{CMCFA}} = \mathbb{E}\left[ \sum_{u,v} \left( \mathcal{L}_{\text{CMCFA}}^{uv} \cdot \exp\{-\text{Var}^{uv}\} + \text{Var}^{uv} \right) \right] \tag{7}$$

**Cross-domain Contrastive Feature Alignment (CDCFA).** CDCFA aligns modality-specific features across source domains using Maximum Mean Discrepancy (MMD) Gretton et al. (2006) to measure distribution differences. The cross-domain contrastive loss for modality $u$ is defined as:

$$\mathcal{L}_{\text{CDCFA}}^{u} = \sum_{s_1, s_2} \sum_{I^{s_1}, I^{s_2}} \left( -\frac{2}{n_{s_1} n_{s_2}} \sum_{i=1}^{n_{s_1}} \sum_{j=1}^{n_{s_2}} k\left( I_i^{s_1}, I_j^{s_2} \right) \right) \tag{8}$$

where $s_1, s_2 \in \text{Dom}$, $\text{Dom} = \{\widetilde{\mathcal{S}}_1, \ldots, \widetilde{\mathcal{S}}_N\}$ or includes $T$ if pseudo-labels are used; $n_{s_1}, n_{s_2}$ denote the batch sizes of domains $s_1$ and $s_2$, respectively; $I_i^s \in X_u^s \cup X_u^{s\prime}$ represents features of the $i$-th sample in domain $s$; and $k$ is a kernel function. For computational efficiency, we adopt a linear kernel $k(x, y) = x^\top y$. The overall CDCFA loss is computed across all $M$ modalities as $\mathcal{L}_{CDCFA} = \sum_{u=1}^{M} \mathcal{L}_{CDCFA}^u$.

**Cross-domain Adversarial Feature Alignment (CDAFA).** CDAFA aligns domains globally in the fused multi-modal feature space by using domain discriminators to separate features from different domains, while encouraging the extractor to produce domain-invariant representations. We use MLB Kim et al. (2016) to implement the multi-modal projection $f_{mm} : \mathcal{X}_1 \times \cdots \times \mathcal{X}_M \to \mathcal{X}_{mm}$, which fuses modality-specific features into a shared space. To achieve global domain alignment in the fused multi-modal feature space, CDAFA introduces a set of domain discriminators $D_{ij}$, each responsible for distinguishing fused features from domain pair $(s_i, s_j)$. Given fused features $f_m^i \in \mathcal{X}_{mm}$ and predicted logits $g_m^i$, a class-conditional projection $G(f_m^i, g_m^i)$ is applied using MultiLinearMap Long et al. (2018). To mitigate overconfidence caused by noisy multi-modal representations, CDAFA further adopt environment label smoothing Zhang et al. (2023) with soft probabilities. The CDAFA loss is formalized as follows:

$$\mathcal{L}_{\text{CDAFA}} = \sum_{s_i, s_j} \left( \mathbb{E}_{x_m^i \sim s_i} W_m^{ij} \log \left[ \alpha + D_{ij}(G(f_m^i, g_m^i)) \right] \right.$$
$$\left. + \mathbb{E}_{x_n^j \sim s_j} W_n^{ij} \log \left[ 1 - \alpha - D_{ij}(G(f_n^j, g_n^j)) \right] \right), \tag{9}$$

where $\alpha = 0.8$ is the smoothing factor. Weights are determined by the entropy of predicted logits as:

$$w_k^{ij} = 1 + \exp\left\{ -g_k^s \cdot \log g_k^s \right\}, \quad W_k^{ij} = \frac{(n_{s_i} + n_{s_j}) \cdot w_k^{ij}}{\sum_{m=1}^{n_{s_i}} w_m^{ij} + \sum_{n=1}^{n_{s_j}} w_n^{ij}}, \tag{10}$$

where $k$ is a sample in domain $s \in \{s_i, s_j\}$, and $n_{s_i}, n_{s_j}$ denote the batch sizes of domains $s_i$ and $s_j$, respectively.

**Uncertainty-aware Classifier Refinement (UACR).** UACR progressively improves the target domain classifier via pseudo-labeling and uncertainty modeling. Specifically, to enable self-learning, a preliminary model is first trained by aligning only the source domains. It then generates target pseudo-labels, which are filtered using uncertainty and confidence from multiple classifiers to reduce noise. For a target sample feature $f^t$, the uncertainty score measures inter-head disagreement among $N$ source-specific classifiers $F_{cls}^i$, and is defined as $s_{uncer} = \exp(-\text{Var}_{ps})$, where $\text{Var}_{ps}$ is the average pairwise KL divergence, formalized as follows:

$$\text{Var}_{ps} = \sum_{i=1}^{N} \sum_{j=i+1}^{N} \left( \mathbb{E}\left[ \text{KL}\left( F_{cls}^i(f^t|\theta_i), F_{cls}^j(f^t|\theta_j) \right) \right] \right.$$
$$\left. + \mathbb{E}\left[ \text{KL}\left( F_{cls}^j(f^t|\theta_j), F_{cls}^i(f^t|\theta_i) \right) \right] \right). \tag{11}$$

Prediction confidence is measured by the aggregated score, obtained by averaging outputs from all $N$ classification heads as follows:

$$s_{cls}^t = \frac{\sum_{i=1}^{N} F_{cls}^i(f^t|\theta_i)}{N}. \tag{12}$$

Pseudo-labels are filtered by the score $= s_{uncer} \cdot s_{cls}^t$, which integrates uncertainty and confidence. Top-ranked samples from each class are selected for reliable self-training.

**Objective Function.** We adopt the standard cross-entropy (CE) loss as classification task loss $\mathcal{L}_{task}$:

$$\mathcal{L}_{task} = \sum_{i=1}^{N} \text{CE}(F_{cls}^i(f_{mm}(X|\theta_i)), y) + \sum_{j=1}^{M} \text{CE}(F_j(X_j|\theta_j), y), \tag{13}$$

where $X = \{X_1, \ldots, X_M\}$ denotes multi-modal features of samples $x \in \widetilde{\mathcal{S}} \cup \mathcal{T}$, and $y$ is the corresponding label or pseudo-label. MCC Jin et al. (2020) is introduced for label-space alignment as $L_{mcc} = MCC(s_{cls}^t)$. The overall objective function is formalized as follows:

$$\mathcal{L}_{M2CAN} = \alpha_1 \cdot \mathcal{L}_{CMCFA} + \beta_1 \cdot \mathcal{L}_{CDCFA} + \gamma \cdot (\mathcal{L}_{CMAFA} + \mathcal{L}_{mcc}) + \mathcal{L}_{task}. \tag{14}$$

where $\alpha_1$, $\beta_1$, and $\gamma$ are hyperparameters used to balance the different loss terms.

## 3.4 THEORETICAL ANALYSIS OF PGPA

**Theorem 1** Let $\mathcal{H}$ be the hypothesis space. Given multiple source domains $\mathcal{S} = \{\mathcal{S}_i\}_{i=1}^{N}$ and target domain $\mathcal{T}$, the expected error on the target domain $R_{\mathcal{T}}(h)$ for hypothesis $h \in \mathcal{H}$ can be bounded by:

$$\forall h \in \mathcal{H}, \ R_{\mathcal{T}}(h) \leq \frac{1}{N} \sum_{i=1}^{N} R_{\mathcal{S}_i}(h) + \frac{1}{2} \sum_{i=1}^{N} d_{\mathcal{H}\Delta\mathcal{H}}(\mathcal{S}_i, \mathcal{T}) + C \tag{15}$$

where $R_{\mathcal{S}_i}(h)$ is the expected source error on the $i$-th source domains, $d_{\mathcal{H}\Delta\mathcal{H}}(R_{\mathcal{S}_i}, \mathcal{T})$ is the $\mathcal{H}\Delta\mathcal{H}$-divergence between the $i$-th source and the target, and $C$ is the shared expected loss term. In conventional MSDA, $C$ is often assumed to be negligibly small and disregarded by methods. However, in the MSM²DA setting, $C$ becomes critical and cannot be ignored due to two key factors. First, different source domains exhibit distinct joint distributions in modalities and labels. This diversity increases the risk of semantic misalignment between source and target domains. Second, in multi-modal learning, the heterogeneity of data distributions across domains affects cross-modal fusion, resulting in poor feature representations.

**Definition 1** $C$ is defined as:

$$C = \min_{h \in \mathcal{H}} \sum_{i=1}^{N} R_{\mathcal{S}_i}(h, f_{\mathcal{S}_i}) + R_{\mathcal{T}}(h, f_{\mathcal{T}}), \tag{16}$$

Let $f_{\mathcal{S}}$ and $f_{\mathcal{T}}$ represent the true labeling functions for the source and target domains, respectively. According to the result in Ben-David et al. (2010), for any pair of labeling functions $f_{\mathcal{S}_i}$ from the source domain and $f_{\mathcal{T}}$ from the target domain, the following inequality holds:

$$R(f_{\mathcal{S}_1}, f_{\mathcal{T}}) \leq R(f_{\mathcal{S}_1}, f_{\mathcal{T}}) + R(f_{\mathcal{S}_2}, f_{\mathcal{T}}) + \cdots + R(f_{\mathcal{S}_N}, f_{\mathcal{T}}) \tag{17}$$

Then, we have:

$$
\begin{aligned}
C &= \min_{h \in \mathcal{H}} \sum_{i=1}^{N} R_{\mathcal{S}}(h, f_{\mathcal{S}_i}) + R_{\mathcal{T}}(h, f_{\mathcal{T}}) \\
&\leq \min_{h \in \mathcal{H}} \sum_{i=1}^{N} R_{\mathcal{S}}(h, f_{\mathcal{S}_i}) + \sum_{i=1}^{N} R_{\mathcal{T}}(h, f_{\mathcal{S}_i}) + \sum_{i=1}^{N} R_{\mathcal{T}}(f_{\mathcal{S}_i}, f_{\mathcal{T}}) \\
&\leq \min_{h \in \mathcal{H}} \sum_{i=1}^{N} R_{\mathcal{S}}(h, f_{\mathcal{S}_i}) + \sum_{i=1}^{N} R_{\mathcal{T}}(h, f_{\mathcal{S}_i}) + \sum_{i=1}^{N} R_{\mathcal{T}}(f_{\mathcal{S}_i}, f_{\hat{\mathcal{T}}}) + \sum_{i=1}^{N} R_{\mathcal{T}}(f_{\mathcal{T}}, f_{\hat{\mathcal{T}}}).
\end{aligned}
\tag{18}
$$

where $f_{\hat{\mathcal{T}}}$ is the pseudo-labeling function. The first two terms measure the disagreement between $h$ and $f_{\mathcal{S}_i}$, which can be minimized by learning $h$ on labeled source data. The third term $R_{\mathcal{T}}(f_{\mathcal{S}_i}, f_{\hat{\mathcal{T}}})$ reflects the discrepancy between $i$-th source and pseudo-label functions, and $R_{\mathcal{T}}(f_{\mathcal{T}}, f_{\hat{\mathcal{T}}})$ is the discrepancy between the true and pseudo-labeling functions in the target domain.

**Reducing Domain Divergence.** PGPA directly contributes to reducing the $\mathcal{H}\Delta\mathcal{H}$-divergence $d_{\mathcal{H}\Delta\mathcal{H}}(\mathcal{S}_i, \mathcal{T})$. By aligning the low-frequency amplitude of the source data with that of the target domain, PGPA effectively narrows the distribution shift between domains, thereby directly optimizing the domain discrepancy term in the error bound.

**Reducing Label Function Discrepancy.** PGPA reduces the discrepancy $R_{\mathcal{T}}(f_{\mathcal{S}_i}, f_{\hat{\mathcal{T}}})$ by aligning source domain semantic structures with target domain perceptual style. The domain-invariant visual inputs enables the feature representations to be learned in a shared space, better regularizing $f_{\hat{\mathcal{T}}}$ to match the source function $f_{\mathcal{S}_i}$ and minimizing their discrepancy. Besides, the visual consistency further facilitates cross-modal alignment, reinforcing the consistency between $f_{\mathcal{S}_i}$ and $f_{\hat{\mathcal{T}}}$.

| Standard | Method | Detail | Avg. | →AVA (→A) | | | | →PCCD (→P) | | | | →RPCD (→R) | | | |
|---|---|---|---|---|---|---|---|---|---|---|---|---|---|---|---|
| | | | | Acc | P | R | F1 | Acc | P | R | F1 | Acc | P | R | F1 |
| Source-only | Single-best | – | 66.3 | 68.0 | 69.1 | 66.6 | 66.3 | 64.7 | 65.3 | 64.4 | 64.1 | 66.2 | 68.9 | 68.6 | 66.2 |
| | Combined | – | 66.7 | 70.5 | 74.3 | 71.9 | 70.1 | 66.1 | 67.4 | 66.0 | 65.4 | 63.5 | 72.3 | 68.0 | 62.7 |
| Single-best DA | CDANLong et al. (2018) | CDAN+ELS | 71.1 | 73.3 | 76.9 | 74.7 | 73.0 | 68.4 | 69.7 | 68.3 | 67.7 | 71.6 | 70.7 | 70.8 | 70.7 |
| | MCCJin et al. (2020) | CDAN+MCC+ELS | 72.6 | 76.0 | 76.6 | 76.6 | 76.0 | 69.2 | 70.7 | 69.1 | 68.6 | 72.7 | 72.7 | 73.4 | 72.5 |
| | SDATRangwani et al. (2022) | CDAN+SDAT+ELS | 70.9 | 77.9 | 77.9 | 78.1 | 77.9 | 68.2 | 69.0 | 68.2 | 67.9 | 66.5 | 67.4 | 66.8 | 66.2 |
| | ELSZhang et al. (2023) | CDAN+MCC+SDAT+ELS | 70.8 | 77.1 | 77.6 | 77.6 | 77.1 | 68.7 | 68.9 | 68.7 | 68.6 | 66.6 | 70.2 | 69.4 | 66.5 |
| | xMUDAJaritz et al. (2020) | Text-only | 72.6 | 75.3 | 76.5 | 76.1 | 75.3 | 69.3 | 71.5 | 69.4 | 68.6 | 73.1 | 73.7 | 74.4 | 73.1 |
| | | Image-only | 54.4 | 54.2 | 60.4 | 50.2 | 35.6 | 50.5 | 63.5 | 50.8 | 35.6 | 58.6 | 29.3 | 50.0 | 36.9 |
| | | Fusion | 72.1 | 74.0 | 77.1 | 72.5 | 72.3 | 69.5 | 71.8 | 69.7 | 69.1 | 72.7 | 73.0 | 73.6 | 72.6 |
| | DsCML Peng et al. (2021) | Text-only | 71.8 | 76.5 | 76.8 | 76.9 | 76.5 | 66.9 | 70.3 | 67.1 | 65.6 | 72.1 | 71.6 | 70.1 | 70.4 |
| | | Image-only | 54.5 | 53.7 | 51.3 | 58.5 | 42.7 | 51.2 | 52.1 | 50.3 | 36.5 | 58.6 | 29.3 | 50.0 | 36.9 |
| | | Fusion | 71.1 | 77.0 | 77.2 | 77.3 | 77.0 | 66.5 | 69.8 | 66.7 | 65.3 | 69.7 | 69.5 | 66.9 | 67.1 |
| Source-combined DA | CDANLong et al. (2018) | CDAN+ELS | 69.3 | 75.7 | 76.3 | 76.2 | 75.7 | 67.5 | 68.1 | 67.5 | 67.2 | 64.8 | 66.0 | 66.3 | 64.8 |
| | MCCJin et al. (2020) | CDAN+MCC+ELS | 71.9 | 77.3 | 77.7 | 76.7 | 76.9 | 67.8 | 68.9 | 67.7 | 67.2 | 70.7 | 73.1 | 73.0 | 70.7 |
| | SDATRangwani et al. (2022) | CDAN+SDAT+ELS | 69.4 | 76.0 | 77.5 | 76.9 | 76.0 | 68.5 | 68.8 | 68.4 | 68.3 | 63.6 | 65.9 | 65.8 | 63.6 |
| | ELSZhang et al. (2023) | CDAN+MCC+SDAT+ELS | 70.7 | 70.4 | 73.5 | 71.7 | 70.0 | 68.8 | 69.3 | 68.9 | 68.7 | 73.0 | 75.3 | 75.3 | 73.0 |
| | xMUDAJaritz et al. (2020) | Text-only | 67.2 | 71.2 | 76.5 | 76.5 | 70.6 | 67.5 | 69.2 | 67.4 | 66.8 | 62.8 | 61.8 | 62.0 | 61.9 |
| | | Image-only | 53.9 | 54.0 | 47.0 | 49.9 | 35.5 | 50.3 | 25.2 | 50.0 | 33.5 | 57.3 | 54.2 | 53.4 | 52.4 |
| | | Fusion | 67.7 | 72.9 | 76.1 | 74.2 | 72.6 | 67.4 | 69.1 | 67.3 | 66.6 | 62.8 | 61.9 | 62.1 | 62.0 |
| | DsCMLPeng et al. (2021) | Text-only | 66.7 | 71.9 | 76.2 | 73.4 | 71.5 | 67.3 | 68.3 | 67.2 | 66.8 | 60.8 | 70.8 | 65.7 | 59.6 |
| | | Image-only | 52.3 | 54.6 | 54.2 | 51.2 | 42.4 | 50.5 | 58.4 | 50.8 | 36.3 | 51.8 | 53.7 | 53.3 | 50.9 |
| | | Fusion | 66.5 | 72.4 | 76.5 | 73.9 | 72.0 | 67.7 | 68.5 | 67.4 | 67.0 | 59.4 | 70.7 | 64.6 | 57.8 |
| MSDA | MDANZhao et al. (2018) | – | 69.8 | 72.9 | 75.8 | 74.1 | 72.6 | 68.5 | 68.5 | 68.5 | 68.4 | 68.1 | 72.9 | 71.4 | 68.0 |
| | M³SDA Peng et al. (2019) | – | 69.8 | 74.9 | 77.3 | 76.0 | 74.7 | 68.0 | 69.5 | 67.9 | 67.3 | 66.5 | 65.3 | 64.6 | 64.8 |
| | T-SVDNet Li et al. (2021) | – | 70.7 | 75.3 | 76.9 | 76.2 | 75.3 | 68.2 | 68.4 | 68.2 | 68.2 | 68.7 | 73.7 | 72.0 | 68.6 |
| MSM²DA | M2CAN Zhao et al. (2025) | – | 74.7 | 79.9 | 79.8 | 80.0 | 79.9 | 69.8 | 69.8 | 69.8 | 69.8 | 74.5 | 74.7 | 75.4 | 74.4 |
| | Ours | – | **75.8** | **81.1** | **81.4** | **81.5** | **81.1** | **70.5** | **70.5** | **70.5** | **70.5** | **75.8** | **76.7** | **77.3** | **75.8** |
| | | Δ | +1.1 | +1.2 | +1.6 | +1.5 | +1.2 | +0.7 | +0.7 | +0.7 | +0.7 | +1.3 | +2.0 | +1.9 | +1.4 |

Table 1: Comparison with state-of-the-art methods on ResNet50+BERT for aesthetics assessment. The best results are highlighted in bold, and the second-best results are underlined. Our method achieves the highest average performance, demonstrating superior cross-domain generalization. The numbers in red indicate the improvement relative to the baseline performance.

| Standard | Method | Detail | Avg. | →TumEmo (→TE) | | | | →T4SA (→T) | | | | →Yelp (→Y) | | | |
|---|---|---|---|---|---|---|---|---|---|---|---|---|---|---|---|
| | | | | Acc | P | R | F1 | Acc | P | R | F1 | Acc | P | R | F1 |
| Source-only | Single-best | – | 58.0 | 57.3 | 59.4 | 57.3 | 57.0 | 61.1 | 61.6 | 61.1 | 56.7 | 55.5 | 37.3 | 55.5 | 44.6 |
| | Combined | – | 56.6 | 56.4 | 58.1 | 56.4 | 56.3 | 58.3 | 59.8 | 58.3 | 58.3 | 55.0 | 52.6 | 55.0 | 48.3 |
| Single-best DA | CDANLong et al. (2018) | CDAN+ELS | 62.7 | 60.9 | 60.4 | 60.9 | 60.5 | 68.5 | 74.9 | 68.5 | 68.9 | 58.7 | 57.5 | 58.7 | 56.8 |
| | MCCJin et al. (2020) | CDAN+MCC+ELS | 61.9 | 61.6 | 60.4 | 61.6 | 60.1 | 67.2 | 67.7 | 67.2 | 67.4 | 56.9 | 56.8 | 56.9 | 56.7 |
| | SDATRangwani et al. (2022) | CDAN+SDAT+ELS | 62.7 | 59.6 | 60.5 | 59.6 | 60.0 | 68.5 | 68.6 | 68.5 | 67.9 | 59.9 | 59.9 | 59.9 | 59.8 |
| | ELSZhang et al. (2023) | CDAN+MCC+SDAT+ELS | 62.3 | 57.5 | 60.0 | 57.5 | 57.4 | 74.1 | 74.7 | 74.1 | 73.9 | 55.3 | 55.1 | 55.3 | 54.6 |
| | xMUDAJaritz et al. (2020) | Text-only | 58.3 | 57.8 | 58.1 | 57.8 | 57.3 | 60.2 | 58.9 | 60.2 | 54.4 | 56.9 | 54.3 | 56.9 | 48.5 |
| | | Image-only | 34.9 | 33.8 | 41.9 | 33.8 | 20.3 | 35.8 | 35.7 | 35.8 | 35.6 | 35.0 | 36.0 | 35.0 | 26.0 |
| | | Fusion | 58.8 | 57.9 | 58.2 | 57.9 | 57.8 | 61.9 | 61.2 | 61.9 | 58.4 | 56.5 | 55.0 | 56.5 | 49.1 |
| | DsCMLPeng et al. (2021) | Text-only | 61.6 | 59.5 | 59.8 | 59.5 | 58.8 | 69.1 | 74.4 | 69.1 | 69.3 | 56.1 | 38.9 | 56.1 | 45.3 |
| | | Image-only | 36.4 | 37.3 | 37.2 | 37.3 | 34.6 | 33.9 | 33.9 | 33.9 | 33.8 | 37.9 | 38.0 | 37.9 | 36.9 |
| | | Fusion | 62.0 | 60.2 | 60.5 | 60.2 | 59.5 | 69.6 | 75.8 | 69.6 | 70.0 | 56.1 | 39.0 | 56.1 | 45.3 |
| Source-combined DA | CDANLong et al. (2018) | CDAN+ELS | 58.9 | 57.9 | 57.4 | 57.9 | 57.6 | 63.0 | 68.7 | 63.0 | 62.0 | 55.8 | 55.1 | 55.8 | 55.1 |
| | MCCJin et al. (2020) | CDAN+MCC+ELS | 62.7 | 57.3 | 56.7 | 57.3 | 55.6 | 75.1 | 78.1 | 75.1 | 75.3 | 55.7 | 55.3 | 55.7 | 55.5 |
| | SDATRangwani et al. (2022) | CDAN+SDAT+ELS | 62.2 | 57.9 | 57.9 | 57.9 | 56.9 | 69.8 | 70.6 | 69.8 | 69.9 | 58.9 | 60.0 | 58.9 | 59.2 |
| | ELSZhang et al. (2023) | CDAN+MCC+SDAT+ELS | 67.9 | 62.3 | 62.8 | 62.3 | 62.3 | 83.4 | 83.6 | 83.4 | 83.5 | 57.9 | 57.7 | 57.9 | 57.2 |
| | xMUDAJaritz et al. (2020) | Text-only | 59.6 | 59.1 | 59.5 | 59.1 | 58.9 | 64.1 | 64.3 | 64.1 | 59.1 | 55.7 | 51.6 | 55.7 | 45.0 |
| | | Image-only | 36.8 | 34.0 | 37.5 | 34.0 | 26.5 | 39.7 | 39.6 | 39.7 | 39.3 | 36.8 | 39.8 | 36.8 | 28.6 |
| | | Fusion | 59.5 | 57.4 | 58.4 | 57.4 | 57.6 | 64.3 | 64.5 | 64.3 | 59.5 | 56.7 | 53.3 | 56.7 | 48.1 |
| | DsCMLPeng et al. (2021) | Text-only | 58.8 | 58.3 | 58.1 | 58.3 | 57.6 | 62.7 | 64.0 | 62.7 | 61.1 | 55.5 | 49.1 | 55.5 | 45.9 |
| | | Image-only | 37.9 | 40.7 | 41.0 | 40.7 | 40.2 | 36.9 | 36.8 | 36.9 | 36.6 | 36.1 | 36.3 | 36.1 | 35.9 |
| | | Fusion | 58.9 | 58.7 | 58.8 | 58.7 | 58.1 | 63.0 | 64.6 | 63.0 | 62.5 | 55.1 | 48.3 | 55.1 | 43.6 |
| MSDA | MDANZhao et al. (2018) | – | 58.8 | 59.1 | 60.1 | 59.1 | 59.2 | 61.9 | 67.8 | 61.9 | 62.2 | 55.5 | 53.1 | 55.5 | 52.8 |
| | M³SDA Peng et al. (2019) | – | 60.4 | 58.0 | 56.7 | 58.0 | 56.9 | 67.1 | 69.9 | 67.1 | 67.1 | 56.1 | 54.7 | 56.1 | 53.6 |
| | T-SVDNet Li et al. (2021) | – | 59.1 | 58.2 | 59.1 | 58.2 | 58.0 | 61.5 | 63.7 | 61.5 | 53.9 | 57.7 | 54.8 | 57.7 | 53.9 |
| MSM²DA | M2CAN Zhao et al. (2025) | – | 69.9 | 63.8 | 63.2 | 63.8 | 63.4 | 84.7 | 84.8 | 84.7 | 84.7 | 61.2 | 61.4 | 61.2 | 61.0 |
| | Ours | – | 71.5 | 64.3 | 63.8 | 64.3 | 64.0 | 86.4 | 86.5 | 86.4 | 86.4 | 63.9 | 64.0 | 63.9 | 64.0 |
| | | Δ | +1.6 | +0.5 | +0.6 | +0.5 | +0.6 | +1.7 | +1.7 | +1.7 | +1.7 | +2.7 | +2.6 | +2.7 | +3.0 |

Table 2: Comparison with state-of-the-art methods for sentiment assessment. The best results are highlighted in bold, and the second-best results are underlined. Our method consistently outperforms others, demonstrating improved sentiment adaptation across domains. The numbers in red indicate the improvement relative to the baseline performance.

## 4 EXPERIMENTS

### 4.1 DATASETS

Following prior work Zhao et al. (2025), we evaluate our method using two groups of datasets, covering aesthetics assessment and sentiment analysis respectively. Each dataset is treated as an individual domain due to differences in data distribution. **For aesthetics assessment**, we use AVA Zhou et al. (2016), PCCD Chang et al. (2017), and RPCD Vera Nieto et al. (2022). For AVA, we label images with an average rating above 5.5 as high-quality and the rest as low-quality. For PCCD, images with a mean score above 8.0 are labeled as high-quality, while others are labeled as low-quality. For RPCD, following Vera Nieto et al. (2022), we retain only samples where both models Liu et al. (2019); Loureiro et al. (2022) yield identical predictions. To ensure a fair comparison across domains, we randomly sample 3,388 images for training and 847 for testing from each dataset. **For sentiment analysis**, we adopt TumEmo Yang et al. (2020), T4SA Vadicamo et al. (2017), and Yelp Truong & Lauw (2019). For TumEmo, emotions are grouped into negative (Angry, Bored, Fear, Sad), neutral (Calm), and positive (Love, Happy). For T4SA, Twitter posters are annotated as negative, neutral, or positive based on content. For Yelp, ratings of 1-2 are labeled negative, 3 as neutral, and 4-5 as positive. To maintain domain balance, we uniformly sample 15,000 training and 1,500 testing examples from each sentiment dataset. In this work, we select one domain as the target and using the remaining domains as sources for MSM²DA task. This results in six adaptation scenarios: AVA (→A), PCCD (→P), and RPCD (→R) for aesthetics-related domains, and TumEmo (→TE), T4SA (→T), and Yelp (→Y) for sentiment-related domains.

### 4.2 COMPARISON WITH THE STATE-OF-THE-ART

We compare our method with four types of baselines, including source-only models, single-source domain adaptation (DA) methods, and multi-source DA methods. Source-only and single-source DA methods are both evaluated under two training settings: single-best, where models are trained on each individual source domain, and source-combined, where models are trained on all source domains jointly. Multi-source DA methods aim to leverage multiple labeled source domains for better generalization to the target domain. We evaluate model performance using five key metrics, including average accuracy across domains (Avg.), domain-specific accuracy (Acc), precision (P), recall (R), and F1-score (F1). All metrics are computed using macro-averaging to ensure fair and balanced evaluation across classes.

As shown in Tables 1 and Figure 2, our method significantly outperforms the baseline methods on both aesthetic and sentiment assessment tasks, achieving consistent improvements across all metrics on their respective three target domains. In terms of average accuracy across domains metirc, our approach achieves gains of 1.1% and 1.6% on the two tasks, respectively, demonstrating strong generalization capability and robustness to imbalanced data. The results confirm that the proposed perceptual alignment strategy effectively enhances cross-modal adaptation and prediction performance under domain shift.

## 4.3 ABLATION STUDY

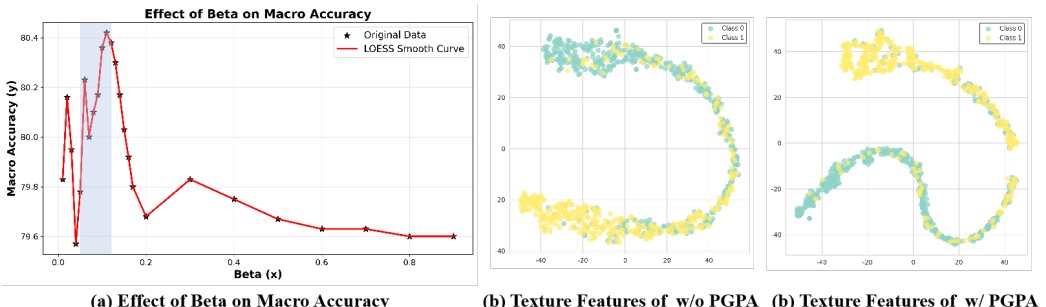

(a) Effect of Beta on Macro Accuracy  (b) Texture Features of w/o PGPA  (b) Texture Features of w/ PGPA

Figure 3: (a) The effect of the parameter $\beta$ on performance, and (b)(c) t-SNE visualizations of text feature distributions for the ablated model (w/o PGPA) and the full model (w/ PGPA), respectively.

**Effect of $\beta$.** As presented in Table 3 (a), we show the effect of various choices of $\beta$ along with the macro accuracy of our method on the AVA dataset. We varied the beta parameter to assess its effect on macro accuracy. Results show the model is sensitive to $\beta$. Specifically, performance shows instability at low $\beta$ values (0.01-0.05),shows an upward trend and reaches the best peak between 0.05-0.12, and declines beyond this range. These results identify 0.05-0.12 as the optimal $\beta$ range for maximizing classification performance, while too small or large values cause degradation.

**Effect on other modality.** To evaluate the impact of PGPA on other modality, we compare text representations under w/o PGPA and w/ PGPA settings. Specifically, we extract the output features from the text feature extracor on the target domain and apply t-SNE under same configurations. As shown in Figure 3 (b), in the w/o PGPA setting, text features exhibit a ring-shaped mixed distribution, where the two classes are interleaved with blurred boundaries, leading to weak discriminability. In contrast, w/ PGPA clearly separates the text features into two independent clusters, as shown in Figure 3 (c), leading to larger inter-class margins and simpler decision boundaries. These findings indicate that applying PGPA to the visual modality significantly enhances the discriminability of text features in the target domain. This improvement may stem from the domain-invariant visual representations constructed by PGPA, which serve as a stable anchor during multimodal interactions and indirectly facilitate the correction of textual feature shifts in the target domain.

## 5 CONCLUSION

In this paper, we propose Phase-Guided Perceptual Alignment (PGPA) to address the low-frequency perceptual discrepancy of the visual modality across multiple domains in Multi-source Multi-modal Domain Adaptation (MSM$^2$DA). PGPA aligns source images to the target domain by replacing their low-frequency amplitude with that of randomly sampled target images in the Fourier domain, while retaining source-phase information. The modified spectrum is then transformed back to the spatial domain, yielding aligned images that preserve semantic structure and reduce perceptual style discrepancy. By performing pixel-level alignment prior to feature extraction, PGPA provides a more stable foundation for subsequent cross-modal fusion and domain adaptation. It is training-free, architecture-independent, and can be seamlessly integrated into existing MSM$^2$DA frameworks. Experimental results on aesthetic assessment and sentiment analysis tasks demonstrate that our method consistently outperforms state-of-the-art approaches, underscoring the importance of perceptual-level alignment in complex cross-domain scenarios.

## 6 ETHICS STATEMENT

This work complies with the ICLR Code of Ethics. We present PGPA, a framework for multi-source multi-modal domain adaptation, evaluated on publicly available benchmark datasets. These datasets contain no personally identifiable or sensitive information, ensuring no risks to privacy or security. Our research advances energy-efficient multi-source multi-modal domain adaptation with potential benefits for scientific and technological applications. All experimental protocols are transparently documented, with fair comparisons to prior work. The contributions are intended solely for research, supporting AI development.

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

# APPENDIX

## A   IMPLEMENTATION DETAILS.

We use a ResNet-50 pre-trained on ImageNet for visual feature extraction and a 12-layer bert-base-uncased BERT model for textual encoding. All classifiers, modality heads, and discriminators are implemented as fully connected layers. Training follows a two-stage strategy. A one-epoch warm-up phase first trains on source domain data only. This is followed by a nine-epoch main phase where filtered target samples with pseudo-labels are gradually incorporated for joint domain alignment. Loss weights are set to 0.5 for domain alignment, 0.2 for modality alignment, and 0.05 for classification. The pseudo-label update rate is fixed at 3. All experiments are implemented in PyTorch and conducted on a single NVIDIA RTX 3090 GPU. We use the Adam optimizer with a batch size of 8. The learning rate is set to 2e-5 for feature extractors and 5e-4 for other modules.

## B   EXAMPLE OF SAMPLE CLASSIFICATION.

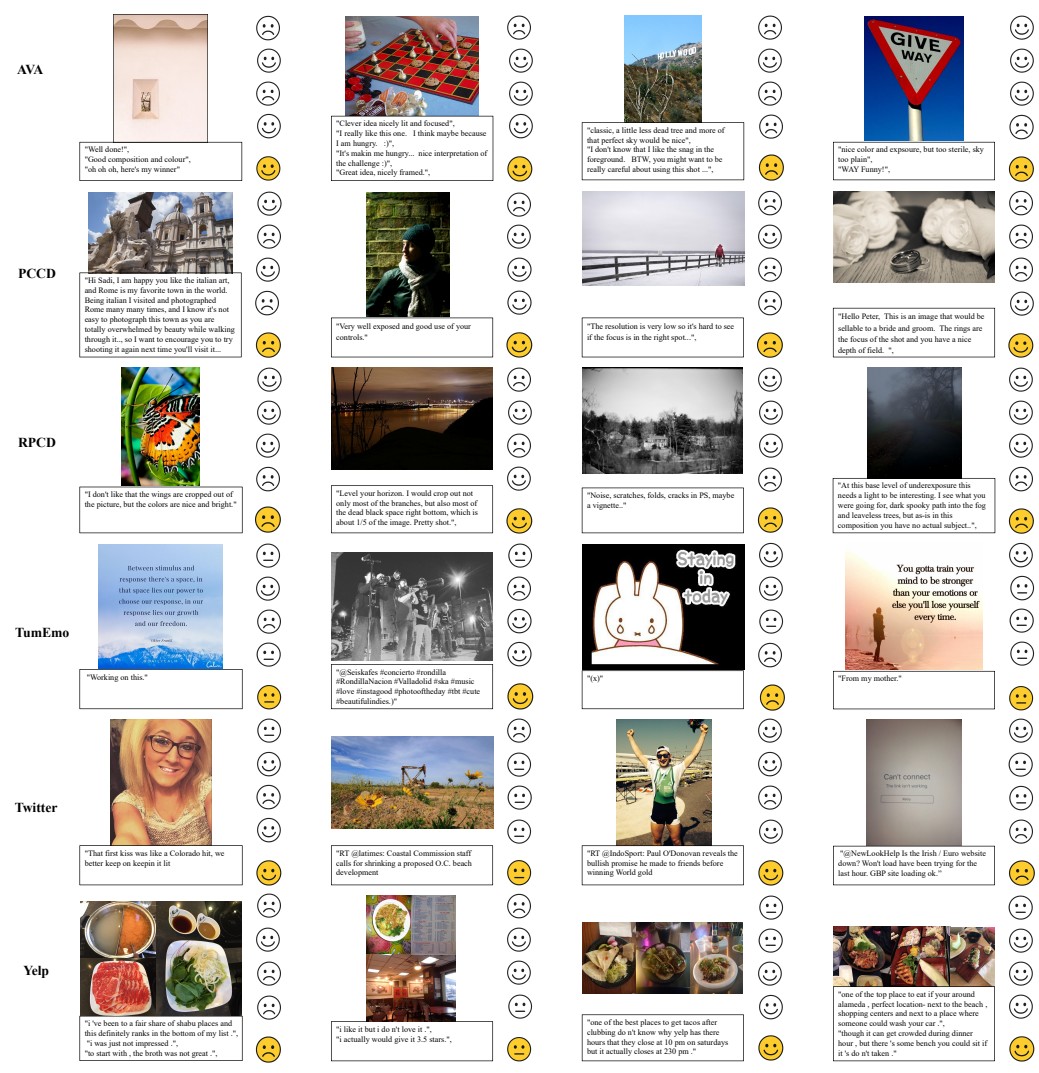

Figure 4: Example results on the aesthetic assessment and sentiment assessment tasks. For each example, predictions from top to bottom are generated by Source-only, MDAN, M2CAN, Ours, and the Ground Truth, respectively.

## C    THE USE OF LARGE LANGUAGE MODELS (LLMs)

Large language models (LLMs) were only used to improve the clarity, grammar, and fluency of the manuscript. They were not involved in the development of research ideas, experimental design, data analysis, or any other aspect of the scientific content.

