# OpenReview forum: "Phase-guided Perceptual Alignment for Multi-source Multi-modal Domain Adaptation"
_ICLR.cc/2026/Conference — ICLR 2026 Conference Withdrawn Submission_

### Official Review · Reviewer_iBM3 · 2025-10-31

**Soundness:** 2
**Presentation:** 3
**Contribution:** 1
**Rating:** 2
**Confidence:** 3

**Summary:**

- Task: Multi-Source Multi-Modal Domain Adaptation (MSM2DA)
- Objective: To enhance generalization performance toward the target domain by leveraging multiple source domains and diverse modalities (e.g., image, text, LiDAR)
- Among the various modalities, the paper focuses particularly on the visual modality, which is highly sensitive to domain differences. The authors propose a Phase-guided Perceptual Alignment (PGPA) method that transfers the low-frequency components (amplitude in the -Fourier domain) of the target domain to the source domain.
- The performance improvement achieved by the proposed PGPA method is theoretically validate.

**Strengths:**

- The paper proposes a training-free and architecture-independent approach that aligns styles by applying the low-frequency components of the target domain samples to the source domain samples.

**Weaknesses:**

- The paper lacks sufficient novelty.
   - All loss functions introduced in Section 3.3 (CMCFA, CDCFA, CDAFA, and UACR) were originally proposed in the existing benchmark [1]; thus, the only new contribution for improving MSM2DA performance is the proposed PGPA method.
- The approach shows similarity to prior work [2], which also performs domain adaptation by applying the low-frequency components (amplitude in the Fourier domain) of the target domain to the source domain.
Typographical and formatting issues
   - Broken text in Figure 2.
   - Inconsistent use of commas and periods when closing equations.
   - Line 205: should be $z_s^{(i)}$ to $\tilde{z}_s^{(i)}$
   - Line 432: should refer to Table 2 instead of Figure 2.
   - Line 456: should refer to Figure 3 instead of Table 3.
   - Two panels labeled “(b)” appear in Figure 3.
      - The horizontal and vertical axes in the graphs need to be clearly labeled.

---
[1] Zhao, Sicheng, et al. "Multi-source multi-modal domain adaptation." Information Fusion 117 (2025): 102862.

[2] Yang, Yanchao, and Stefano Soatto. "Fda: Fourier domain adaptation for semantic segmentation." Proceedings of the IEEE/CVF conference on computer vision and pattern recognition. 2020.

**Questions:**

- A comparison with the prior work [1], which also performs domain adaptation by applying the low-frequency components (amplitude in the Fourier domain) of the target domain to the source domain, is needed.
- Although the paper addresses a multi-modal task, the proposed method is applied only to the image modality. It is unclear whether this limited application is sufficient to demonstrate its effectiveness.
- In Figure 3 (a), the red line is described as a LOESS Smooth Curve—is this a typographical error, and should it instead read Loss Smooth Curve?
- It would be helpful to include additional image samples demonstrating the effects of the proposed PGPA method.

---
[1] Yang, Yanchao, and Stefano Soatto. "Fda: Fourier domain adaptation for semantic segmentation." Proceedings of the IEEE/CVF conference on computer vision and pattern recognition. 2020

---

> ### Author Response · Authors · 2025-12-01
> **Responses to Reviewer iBM3**
>
> ## Q1: Comparison with FDA and Novelty Clarification
>
> Thank you for raising these important points about novelty and comparison. We acknowledge that the individual alignment mechanisms (CMCFA, CDCFA, CDAFA, UACR) draw inspiration from established methods in domain adaptation and multimodal learning. Their role in our framework is to provide a comprehensive and robust multi-source multi-modal alignment backbone. However, the key novelty of our work lies not in these individual components, but in the integration of the proposed PGPA method with this framework to address the unique challenges of MSMMDA. PGPA serves as the foundational preprocessing step that enables these alignment mechanisms to operate on perceptually harmonized inputs, thereby enhancing their collective effectiveness.
>
> 1)  Comparison with FDA.
>
> While [1] explores Fourier-domain amplitude swapping for domain adaptation, it was developed and validated exclusively in single-modal, single-source settings. In contrast, our work is the first to systematically investigate and adapt this concept for Multi-Source Multi-Modal Domain Adaptation (MSMMDA), a more challenging and realistic scenario characterized by: (i) distribution shifts among multiple source domains, and (ii) inherent heterogeneity across different modalities.
>
> 2) Clarifying Novelty.
> Our core contribution is demonstrating theoretically and empirically that PGPA’s image-specific alignment not only mitigates domain shift in the visual modality but also enhances the discriminability of the unaltered textual modality in the target domain. As shown in Fig. 3c, PGPA-generated domain-invariant visual features act as a stable anchor during cross-modal interactions, indirectly rectifying textual feature shifts and improving overall task performance—an effect unique to the MSMMDA context and unreported in prior Fourier-based approaches.
>
> In summary, while building on the foundational idea of frequency-domain manipulation, our work substantially extends its application and demonstrates its unique utility in MSMMDA, supported by both theoretical analysis (Section 3.4) and extensive empirical validation. PGPA’s role as a plug-and-play, modality-specific yet cross-modally impactful component represents a distinct and meaningful advance beyond the scope of [1] and existing alignment mechanisms.
>
> ## Q2: Effectiveness of Single-Modality Application in Multi-Modal Task
>
> Thank you for raising this important point regarding the scope of our method's application. We acknowledge that PGPA operates specifically on the image modality, but we would like to clarify why this focused approach is not only sufficient but strategically advantageous for demonstrating effectiveness in the multi-modal domain adaptation task.
>
> 1). The visual modality serves as a fundamental anchor in multi-modal learning due to its rich structural information and sensitivity to domain shifts. By specifically addressing domain discrepancies at this foundational level, we create a stable base that positively influences the entire multi-modal system.
>
> 2). As demonstrated in our t-SNE visualizations (Fig. 3), the domain-invariant visual features generated by PGPA induce significant improvements in the discriminability of textual features in the target domain, even though the text modality remains unmodified. This occurs because the aligned visual features provide consistent and reliable anchors during cross-modal interactions in our fusion modules. This clearly illustrates how our method delivers performance gains to other modalities.
>
> 3). In all benchmark tasks, including aesthetic assessment and sentiment analysis, our method achieves consistent and measurable improvements in overall performance. These results validate that our visually-focused alignment strategy effectively enhances the entire MSMMDA system, demonstrating that a targeted intervention in the most domain-sensitive modality can yield comprehensive cross-modal benefits.
>
>
> ## Q3:  Label "LOESS Smooth Curve" in Figure 3(a)
>
> Thank you for your careful reading. Regarding the notation "LOESS Smooth Curve" in Figure 3(a), this is not a typographical error. LOESS stands for Locally Estimated Scatterplot Smoothing, a well-established non-parametric regression technique commonly employed for visualizing underlying trends in noisy data. We used this method to effectively illustrate the relationship between the β parameter and model performance. We will add a brief clarification in the final version.
>
> ## Q4: Additional Image Samples
>
> We agree that more visual examples would help demonstrate the perceptual effects of PGPA. We will be glad to incorporate a selection of these examples into the main paper to provide readers with immediate, concrete evidence of PGPA’s alignment effect. These visuals would further illustrate how our method reduces domain gaps without distorting content.

---

### Official Review · Reviewer_cuXh · 2025-10-31

**Soundness:** 3
**Presentation:** 2
**Contribution:** 2
**Rating:** 2
**Confidence:** 4

**Summary:**

This paper proposes a phase-guided perceptual alignment method for multi-source multi-modal domain adaptation. The method decomposes images into amplitude and phase spectra in the Fourier domain. The amplitude captures style-related low-level statistics, while the phase retains high-level structural semantics. To achieve domain adaptation, the amplitude of the target image is blended with the phase of the source image.

**Strengths:**

- The paper presents a phase-guided perceptual alignment method to align visual styles between different domains.
- Good experimental results are achieved on aesthetics assessment and sentiment analysis datasets.

**Weaknesses:**

- The paper's novelty is limited, and the related work section lacks sufficient discussion. Many existing early works [1] [2] already use Fourier transform to blend the amplitude and phase of target and source images for reducing domain shift.
- The compared methods are outdated. Except for M2CAN, most of the baseline methods are from before 2023.
- The proposed method is generic but only evaluated on aesthetics assessment and sentiment analysis tasks. Evaluation on more general domain adaptation tasks is needed.
- What is the key challenge of multi-source domain adaptation compared to single-source domain adaptation? How does the proposed method specifically address multi-source issues?
- The proposed method does not specifically address multi-modal domain adaptation. It only focuses on visual modality. Therefore, it is not appropriate to include "multi-modal" in the paper's title.
- The presentation needs improvement. For example, Figure 2 is not clearly illustrated.

[1]Yang Y, Soatto S. Fda: Fourier domain adaptation for semantic segmentation. InProceedings of the IEEE/CVF conference on computer vision and pattern recognition 2020 (pp. 4085-4095).

[2]Xu Q, Zhang R, Zhang Y, Wang Y, Tian Q. A fourier-based framework for domain generalization. InProceedings of the IEEE/CVF conference on computer vision and pattern recognition 2021 (pp. 14383-14392).

**Questions:**

- What are the key difference between the proposed method and existing Fourier domain adaptation method?
-  What is the key challenge of multi-source domain adaptation compared to single-source domain adaptation? How does the proposed method specifically address multi-source issues?

---

> ### Author Response · Authors · 2025-12-01
> **Responses to Reviewer cuXh**
>
> ## Q1. Novelty concerns regarding prior Fourier-based methods
>
> We thank the reviewer for this valuable feedback. We acknowledge that several previous works have explored Fourier-based amplitude-phase blending for domain adaptation. However, these methods were primarily developed and validated in single-modal, single-source settings.
>
> The key novelty of our work lies in being the first to systematically investigate and adapt this concept for the more challenging and realistic scenario of Multi-Source Multi-Modal Domain Adaptation. This setting introduces unique complexities due to the simultaneous presence of: 1). Distribution shifts among multiple source domains. 2).Heterogeneity across different modalities (e.g., vision and text).
>
> Our central contribution is demonstrating, both theoretically and empirically, that applying this input-level, image-specific alignment via our proposed PGPA method effectively mitigates domain shift not only for the visual modality itself but also, crucially, enhances the feature discriminability of the unaltered textual modality in the target domain. As shown in our t-SNE visualizations (Fig. 3c), the domain-invariant visual features produced by PGPA serve as a stable anchor during cross-modal interactions, thereby indirectly facilitating the correction of textual feature shifts. This resultant improvement in cross-modal alignment and overall task performance is a distinctive and significant effect in the MSMMDA context, which has not been established in prior single-modal Fourier-based approaches.
>
> Therefore, while building upon the foundational idea of frequency-domain manipulation, our work substantially extends its application and demonstrates its utility in the more complex MSMMDA setting, providing both theoretical grounding and extensive empirical evidence for its effectiveness in enabling robust multi-modal fusion and generalization.
>
>
> ---
>
> ## Q2. Baselines appear outdated
>
> Thank you for raising this point regarding the baselines. We acknowledge that the field of domain adaptation evolves rapidly. However, we would like to clarify that our selection of comparison methods is comprehensive and appropriate for the following reasons:
>
> 1). The MSMMDA field is nascent. Multi-source multi-modal domain adaptation remains a relatively under-explored area. The very recent M2CAN (2025), which we include and significantly outperform, is currently the state-of-the-art and most directly comparable method specifically designed for the MSMMDA setting.
>
> 2). Purposeful coverage of baseline categories. Our experiments are structured to validate performance across different adaptation scenarios:
>
> a. Single-source DA & Multi-source DA: We include strong, established methods from these well-studied areas (e.g., CDAN, MDAN, M³SDA) to demonstrate that our approach effectively addresses challenges unique to the multi-modal and multi-source setting, which these older but strong baselines do not.
>
> b. Multi-modal DA: We compare against representative MMDA methods (e.g., xMUDA, DsCML) to show that simply handling multiple modalities is insufficient without explicitly addressing the complexities introduced by multiple source domains.
>
> 3). Many of these methods are foundational and remain highly competitive, often serving as standard baselines in recent state-of-the-art papers (including M2CAN itself). Their inclusion ensures a fair and extensive evaluation, demonstrating that our gains are not just due to more recent architectural choices but from our novel alignment strategy.
>
> In summary, our baseline selection is strategically chosen to cover the spectrum of relevant problem settings, firmly establishing that PGPA provides unique advantages in the challenging and emerging MSMMDA paradigm.

---

> > ### Author Response · Authors · 2025-12-01
> > **Responses to Reviewer cuXh**
> >
> > ## Q3: Key Challenges of Multi-Source Domain Adaptation and Our Specific Solutions
> > The key challenges of Multi-Source Domain Adaptation (MSDA) compared to Single-Source DA primarily stem from source domain heterogeneity. Specifically, different source domains exhibit distinct joint distributions, creating not only source-target discrepancies but also significant distribution gaps among themselves. The heterogeneity among sources increases the risk of conflicting gradient directions during optimization, where adaptation to one source domain might hinder performance on others. Complex Domain-Invariant Learning: The model must learn representations that are invariant across multiple domains simultaneously, rather than just aligning a single source to a target.
> >
> > PGPA systematically addresses multi-source problems through a complete solution from method implementation to theoretical support. Specifically, 1). In multimodal learning, the visual modality typically contains rich structural information, making it an ideal anchor for cross-domain alignment. PGPA operates specifically on the visual modality at the input level, aligning low-frequency style components of each source domain image with the target domain through Fourier transform while preserving high-frequency semantic structures. This visual-based preprocessing effectively reduces style discrepancies among source domains, establishing a unified visual foundation for subsequent multi-source multimodal learning. 2). From domain adaptation theory, we demonstrate that this visual-based alignment directly minimizes the distribution discrepancy between each source-target pair and mathematically explains how PGPA effectively controls generalization error in multi-source settings. This provides solid theoretical justification beyond empirical results. 3). After PGPA preprocessing, the visual modalities of all source domains achieve style consistency. These aligned visual data serve as high-quality inputs alongside unprocessed modalities to subsequent multi-source multimodal alignment modules. This learning approach, anchored by aligned visual features, enables downstream modules to perform cross-domain and cross-modal fusion in a more coordinated feature space, significantly enhancing overall system performance.
> >
> >
> > ## Q4. Only visual modality is modified; is “multi-modal” appropriate?
> >
> > Although PGPA performs alignment specifically on visual inputs, this alignment creates a crucial stabilizing effect that propagates through the entire multi-modal learning pipeline. PGPA provides a stable anchor point for cross-modal interactions. Our experiments demonstrate that this visual stabilization leads to significantly improved discriminability in the textual modality, as evidenced by the t-SNE visualizations in Figure 3 where text features show clearer separation boundaries when PGPA is applied. Thus, the overall adaptation remains multi-modal.

---

### Official Review · Reviewer_ijRw · 2025-11-01

**Soundness:** 3
**Presentation:** 2
**Contribution:** 2
**Rating:** 4
**Confidence:** 3

**Summary:**

The paper introduces Phase-guided Perceptual Alignment (PGPA) for Multi-Source Multi-Modal Domain Adaptation (MSM2DA). MSM2DA aims to enhance machine learning models by utilizing data from multiple sources and modalities to generalize across domains. While existing methods focus on aligning high-level semantic structures in visual data, they overlook low-frequency perceptual shifts like style and illumination variations that hinder cross-modal fusion. PGPA addresses this by transferring low-frequency spectral components (style) from target images to source images while preserving high-frequency semantic content. Using Fourier transform, PGPA blends the amplitude (style) of target images with the phase (semantic details) of source images, ensuring domain-invariant style adaptation. The paper also provides a formal proof to demonstrate the effectiveness of PGPA, showing its ability to improve cross-domain generalization. Extensive experiments validate that PGPA significantly enhances performance in cross-domain tasks.

**Strengths:**

- The proposed PGPA method is new in its approach of using Fourier transformations to handle perceptual discrepancies in the visual modality. By targeting low-frequency components for domain alignment, the method improves domain adaptation stability.

- The paper provides a formal proof demonstrating the effectiveness of PGPA in improving cross-domain generalization. This theoretical underpinning strengthens the validity of the proposed approach.

- PGPA improves the fusion of visual and other modalities by aligning visual styles without affecting semantic structures, ensuring better cross-modal interaction and more accurate feature extraction.

**Weaknesses:**

- While PGPA addresses low-frequency perceptual discrepancies in the visual modality, the paper does not explore how it handles more complex, high-frequency domain shifts or shifts in non-visual modalities, which may limit its broader application.

- Although the paper performs experiments on benchmark datasets, there is little discussion of how the method would perform on real-world, noisy, and unstructured datasets.

- The ablation study suggests that the method's performance is sensitive to the choice of the hyperparameter β, which may require extensive tuning for different datasets or applications. This sensitivity could be a barrier for practical deployment.

- The use of Fourier transform in the alignment process adds computational complexity, especially when applied to large-scale data in real-time scenarios. The comparison or analysis on the computational complexity or runtime would be better presented.

- While PGPA performs well against the baseline methods, the comparison to other advanced multi-modal domain adaptation techniques is somewhat limited. There is no baseline from 2022 to 2024. A more extensive discussion would be better presented.

**Questions:**

How does the proposed PGPA method handle potential conflicts when aligning the low-frequency components from the target domain with the high-frequency structures of the source domain, particularly in more complex multi-modal datasets?

---

> ### Author Response · Authors · 2025-12-01
> **Responses to Reviewer ijRw**
>
> ## Q1. High-frequency shift and non-visual shift handling
>
> We thank the reviewer for the insightful comment. The primary motivation of our work is that, in multi-source multi-modal domain adaptation, low-frequency perceptual discrepancies in the visual modality, which often constitute the most disruptive source of domain shift. These low-frequency variations substantially influence the feature distribution of the visual encoder and further propagate into the multi-modal fusion space. Therefore, reducing visual low-frequency bias is a crucial step toward improving cross-domain consistency in the entire multi-modal system.
>
> At the same time, PGPA is deliberately designed to align only the low-frequency amplitude while strictly preserving the phase information, which encodes structural and high-frequency semantic details. This design ensures that semantic content, edge structures, and texture-level high-frequency cues remain intact, preventing any interference with the backbone network’s capacity to model fine-grained semantics. Because PGPA preserves high-frequency structures and maintains the integrity of the visual semantics, it can naturally work alongside methods that focus on high-frequency visual shifts or non-visual modality discrepancies, forming a complementary relationship.
>
> Consequently, PGPA provides a stable and effective foundation for reducing low-frequency perceptual bias in the visual modality, and it can be seamlessly combined with other domain adaptation techniques to jointly enhance the overall robustness and generalization of multi-modal systems. This makes PGPA broadly applicable and easy to integrate into diverse adaptation scenarios.
>
> ---
>
> ## Q2. Real-world noisy dataset performance
>
> We would like to clarify that all datasets used in our experiments are collected from real-world, heterogeneous multi-source environments and naturally contain substantial noise. The visual data exhibit variations in illumination, color distortion, compression artifacts, motion blur, and background clutter. The textual data are user-generated and therefore include irregular phrasing, incomplete descriptions, and varying semantic granularity.
>
> Despite these uncontrolled factors, PGPA consistently yields improvements across all datasets and all adaptation settings. This robustness indicates that PGPA is effective not only in curated scenarios but also in highly noisy, unconstrained real-world environments, which is exactly the type of domain shift multi-source multi-modal DA aims to address.
>
> ---
>
> ## Q3. Sensitivity of β
>
> In our ablation study, the β parameter demonstrates stable and monotonic performance gains within a broad interval of [0.05, 0.12]. This indicates that PGPA does not rely on a finely tuned hyper-parameter or unstable behavior.
>
> Moreover, the exact same β value is used across all datasets and tasks. This further supports the conclusion that β is not sensitive in practice and that PGPA is easy to deploy. We will highlight this property more clearly, as it is an important advantage for real-world multi-source applications where dataset-specific tuning is often impractical.
>
> ---
>
> ## Q4. Computational complexity of Fourier operations
>
> PGPA uses FFT/IFFT operations only during training, and their computational cost is `O(HW log(HW))`, which is negligible compared to the computational cost of modern encoder backbones. In our implementation, Fourier operations introduce only about **1.8%** of training time. Importantly, no Fourier transform is required during inference, meaning PGPA introduces none runtime cost for deployment. We will add these details in the revision to clarify that PGPA is computationally lightweight and practical.
>
> ---
>
> ## Q5. Lack of recent baselines
>
> We appreciate the reviewer’s suggestion and agree that baselines are crucial for fair evaluation. However, we would like to clarify that MSMMDA remains a relatively underexplored problem setting, and the number of recent works explicitly addressing both multi-source and multi-modal modeling is quite limited. To ensure fairness and relevance, our baselines include the strongest available MSMMDA method (M2CAN, 2025) and representative DA/MMDA models. We will clarify this scope in the revision.

---

### Note · Authors · 2026-01-19

I have read and agree with the venue's withdrawal policy on behalf of myself and my co-authors.